# Autoregressive Score Matching

**Chenlin Meng**
Stanford University
chenlin@stanford.edu

**Lantao Yu**
Stanford University
lantaoyu@cs.stanford.edu

**Yang Song**
Stanford University
yangsong@cs.stanford.edu

**Jiaming Song**
Stanford University
tsong@cs.stanford.edu

**Stefano Ermon**
Stanford University
ermon@cs.stanford.edu

## Abstract

Autoregressive models use chain rule to define a joint probability distribution as a product of conditionals. These conditionals need to be normalized, imposing constraints on the functional families that can be used. To increase flexibility, we propose autoregressive conditional score models (AR-CSM) where we parameterize the joint distribution in terms of the derivatives of univariate log-conditionals (scores), which need not be normalized. To train AR-CSM, we introduce a new divergence between distributions named Composite Score Matching (CSM). For AR-CSM models, this divergence between data and model distributions can be computed and optimized efficiently, requiring no expensive sampling or adversarial training. Compared to previous score matching algorithms, our method is more scalable to high dimensional data and more stable to optimize. We show with extensive experimental results that it can be applied to density estimation on synthetic data, image generation, image denoising, and training latent variable models with implicit encoders.

## 1 Introduction

Autoregressive models play a crucial role in modeling high-dimensional probability distributions. They have been successfully used to generate realistic images [18, 21], high-quality speech [17], and complex decisions in games [29]. An autoregressive model defines a probability density as a product of conditionals using the chain rule. Although this factorization is fully general, autoregressive models typically rely on simple probability density functions for the conditionals (e.g. a Gaussian or a mixture of logistics) [21] in the continuous case, which limits the expressiveness of the model.

To improve flexibility, energy-based models (EBM) represent a density in terms of an energy function, which does not need to be normalized. This enables more flexible neural network architectures, but requires new training strategies, since maximum likelihood estimation (MLE) is intractable due to the normalization constant (partition function). Score matching (SM) [9] trains EBMs by minimizing the Fisher divergence (instead of KL divergence as in MLE) between model and data distributions. It compares distributions in terms of their log-likelihood gradients (scores) and completely circumvents the intractable partition function. However, score matching requires computing the trace of the Hessian matrix of the model's log-density, which is expensive for high-dimensional data [14].

To avoid calculating the partition function without losing scalability in high dimensional settings, we leverage the chain rule to decompose a high dimensional distribution matching problem into simpler univariate sub-problems. Specifically, we propose a new divergence between distributions, named Composite Score Matching (CSM), which depends only on the derivatives of *univariate* log-conditionals (scores) of the model, instead of the full gradient as in score matching. CSM training is

particularly efficient when the model is represented directly in terms of these univariate conditional scores. This is similar to a traditional autoregressive model, but with the advantage that conditional scores, unlike conditional distributions, do not need to be normalized. Similar to EBMs, removing the normalization constraint increases the flexibility of model families that can be used.

Leveraging existing and well-established autoregressive models, we design architectures where we can evaluate all dimensions in parallel for efficient training. During training, our CSM divergence can be optimized directly without the need of approximations [15, 25], surrogate losses [11], adversarial training [5] or extra sampling [3]. We show with extensive experimental results that our method can be used for density estimation, data generation, image denoising and anomaly detection. We also illustrate that CSM can provide accurate score estimation required for variational inference with implicit distributions [8, 25] by providing better likelihoods and FID [7] scores compared to other training methods on image datasets.

## 2   Background

Given i.i.d. samples $\{\mathbf{x}^{(1)}, ..., \mathbf{x}^{(N)}\} \subset \mathbb{R}^D$ from some unknown data distribution $p(\mathbf{x})$, we want to learn an unnormalized density $\tilde{q}_\theta(\mathbf{x})$ as a parametric approximation to $p(\mathbf{x})$. The unnormalized $\tilde{q}_\theta(\mathbf{x})$ uniquely defines the following normalized probability density:

$$q_\theta(\mathbf{x}) = \frac{\tilde{q}_\theta(\mathbf{x})}{Z(\theta)}, \ Z(\theta) = \int \tilde{q}_\theta(\mathbf{x}) d\mathbf{x}, \tag{1}$$

where $Z(\theta)$, the partition function, is generally intractable.

### 2.1   Autoregressive Energy Machine

To learn an unnormalized probabilistic model, [15] proposes to approximate the normalizing constant using one dimensional importance sampling. Specifically, let $\mathbf{x} = (x_1, ..., x_D) \in \mathbb{R}^D$. They first learn a set of one dimensional conditional energies $E_\theta(x_d|\mathbf{x}_{<d}) \triangleq -\log \tilde{q}_\theta(x_d|\mathbf{x}_{<d})$, and then approximate the normalizing constants using importance sampling, which introduces an additional network to parameterize the proposal distribution. Once the partition function is approximated, they normalize the density to enable maximum likelihood training. However, approximating the partition function not only introduces bias into optimization but also requires extra computation and memory usage, lowering the training efficiency.

### 2.2   Score Matching

To avoid computing $Z(\theta)$, we can take the logarithm on both sides of Eq. (1) and obtain $\log q_\theta(\mathbf{x}) = \log \tilde{q}_\theta(\mathbf{x}) - \log Z(\theta)$. Since $Z(\theta)$ does not depend on $\mathbf{x}$, we can ignore the intractable partition function $Z(\theta)$ when optimizing $\nabla_\mathbf{x} \log q_\theta(\mathbf{x})$. In general, $\nabla_\mathbf{x} \log q_\theta(\mathbf{x})$ and $\nabla_\mathbf{x} \log p(\mathbf{x})$ are called the *score* of $q_\theta(\mathbf{x})$ and $p(\mathbf{x})$ respectively. Score matching (SM) [9] learns $q_\theta(\mathbf{x})$ by matching the scores between $q_\theta(\mathbf{x})$ and $p(\mathbf{x})$ using the Fisher divergence:

$$L(q_\theta; p) \triangleq \frac{1}{2} \mathbb{E}_p[\|\nabla_\mathbf{x} \log p(\mathbf{x}) - \nabla_\mathbf{x} \log q_\theta(\mathbf{x})\|_2^2]. \tag{2}$$

Ref. [9] shows that under certain regularity conditions $L(\theta; p) = J(\theta; p) + C$, where $C$ is a constant that does not depend on $\theta$ and $J(\theta; p)$ is defined as below:

$$J(\theta; p) \triangleq \mathbb{E}_p \left[ \frac{1}{2} \|\nabla_\mathbf{x} \log q_\theta(\mathbf{x})\|_2^2 + \mathrm{tr}(\nabla_\mathbf{x}^2 \log q_\theta(\mathbf{x})) \right],$$

where $\mathrm{tr}(\cdot)$ denotes the trace of a matrix. The above objective does not involve the intractable term $\nabla_\mathbf{x} \log p(\mathbf{x})$. However, computing $\mathrm{tr}(\nabla_\mathbf{x}^2 \log q_\theta(\mathbf{x}))$ is in general expensive for high dimensional data. Given $\mathbf{x} \in \mathbb{R}^D$, a naive approach requires $D$ times more backward passes than computing the gradient $\nabla_\mathbf{x} \log q_\theta(\mathbf{x})$ [25] in order to compute $\mathrm{tr}(\nabla_\mathbf{x}^2 \log q_\theta(\mathbf{x}))$, which is inefficient when $D$ is large. In fact, ref. [14] shows that within a constant number of forward and backward passes, it is unlikely for an algorithm to be able to compute the diagonal of a Hessian matrix defined by any arbitrary computation graph.

# 3 Composite Score Matching

To make SM more scalable, we introduce Composite Score Matching (CSM), a new divergence suitable for learning unnormalized statistical models. We can factorize any given data distribution $p(\mathbf{x})$ and model distribution $q_\theta(\mathbf{x})$ using the chain rule according to a common variable ordering:

$$p(\mathbf{x}) = \prod_{d=1}^{D} p(x_d|\mathbf{x}_{<d}), \quad q_\theta(\mathbf{x}) = \prod_{d=1}^{D} q_\theta(x_d|\mathbf{x}_{<d})$$

where $x_d \in \mathbb{R}$ stands for the $d$-th component of $\mathbf{x}$, and $\mathbf{x}_{<d}$ refers to all the entries with indices smaller than $d$ in $\mathbf{x}$. Our key insight is that instead of directly matching the joint distributions, we can match the conditionals of the model $q_\theta(x_d|\mathbf{x}_{<d})$ to the conditionals of the data $p(x_d|\mathbf{x}_{<d})$ using the Fisher divergence. This decomposition results in simpler problems, which can be optimized efficiently using *one-dimensional* score matching. For convenience, we denote the conditional scores of $q_\theta$ and $p$ as $s_{\theta,d}(\mathbf{x}_{<d}, x_d) \triangleq \frac{\partial}{\partial x_d} \log q_\theta(x_d|\mathbf{x}_{<d}) : \mathbb{R}^{d-1} \times \mathbb{R} \to \mathbb{R}$ and $s_d(\mathbf{x}_{<d}, x_d) \triangleq \frac{\partial}{\partial x_d} \log p(x_d|\mathbf{x}_{<d}) : \mathbb{R}^{d-1} \times \mathbb{R} \to \mathbb{R}$ respectively. This gives us a new divergence termed *Composite Score Matching (CSM)*:

$$L_{CSM}(q_\theta; p) = \frac{1}{2} \sum_{d=1}^{D} \mathbb{E}_{p(\mathbf{x}_{<d})} \mathbb{E}_{p(x_d|\mathbf{x}_{<d})} \left[ (s_d(\mathbf{x}_{<d}, x_d) - s_{\theta,d}(\mathbf{x}_{<d}, x_d))^2 \right]. \quad (3)$$

This divergence is inspired by composite scoring rules [1], a general technique to decompose distribution-matching problems into lower-dimensional ones. As such, it bears some similarity with pseudo-likelihood, a composite scoring rule based on KL-divergence. As shown in the following theorem, it can be used as a learning objective to compare probability distributions:

**Theorem 1** (CSM Divergence). $L_{CSM}(q_\theta, p)$ *vanishes if and only if* $q_\theta(\mathbf{x}) = p(\mathbf{x})$ *a.e.*

*Proof Sketch.* If the distributions match, their derivatives (conditional scores) must be the same, hence $L_{CSM}$ is zero. If $L_{CSM}$ is zero, the conditional scores must be the same, and that uniquely determines the joints. See Appendix for a formal proof. □

Eq. (3) involves $s_d(\mathbf{x})$, the unknown score function of the data distribution. Similar to score matching, we can apply integration by parts to obtain an equivalent but tractable expression:

$$J_{CSM}(\theta; p) = \sum_{d=1}^{D} \mathbb{E}_{p(\mathbf{x}_{<d})} \mathbb{E}_{p(x_d|\mathbf{x}_{<d})} \left[ \frac{1}{2} s_{\theta,d}(\mathbf{x}_{<d}, x_d)^2 + \frac{\partial}{\partial x_d} s_{\theta,d}(\mathbf{x}_{<d}, x_d) \right], \quad (4)$$

The equivalence can be summarized using the following results:

**Theorem 2** (Informal). *Under some regularity conditions,* $L_{CSM}(\theta; p) = J_{CSM}(\theta; p) + C$ *where* $C$ *is a constant that does not depend on* $\theta$.

*Proof Sketch.* Integrate by parts the one-dimensional SM objectives. See Appendix for a proof. □

**Corollary 1.** *Under some regularity conditions,* $J_{CSM}(\theta, p)$ *is minimized when* $q_\theta(\mathbf{x}) = p(\mathbf{x})$ *a.e.*

In practice, the expectation in $J_{CSM}(\theta; p)$ can be approximated by a sample average using the following unbiased estimator

$$\hat{J}_{CSM}(\theta; p) \triangleq \frac{1}{N} \sum_{i=1}^{N} \sum_{d=1}^{D} \left[ \frac{1}{2} s_{\theta,d}(\mathbf{x}_{<d}^{(i)}, x_d^{(i)})^2 + \frac{\partial}{\partial x_d^{(i)}} s_{\theta,d}(\mathbf{x}_{<d}^{(i)}, x_d^{(i)}) \right], \quad (5)$$

where $\{\mathbf{x}^{(1)}, ..., \mathbf{x}^{(N)}\}$ are i.i.d samples from $p(\mathbf{x})$. It is clear from Eq. (5) that evaluating $\hat{J}_{CSM}(\theta; p)$ is efficient as long as it is efficient to evaluate $s_{\theta,d}(\mathbf{x}_{<d}, x_d) \triangleq \frac{\partial}{\partial x_d} \log q_\theta(x_d|\mathbf{x}_{<d})$ and its derivative $\frac{\partial}{\partial x_d} s_{\theta,d}(\mathbf{x}_{<d}, x_d)$. This in turn depends on how the model $q_\theta$ is represented. For example, if $q_\theta$ is an energy-based model defined in terms of an energy $\tilde{q}_\theta$ as in Eq. (1), computing $q_\theta(x_d|\mathbf{x}_{<d})$ (and hence its derivative, $s_{\theta,d}(\mathbf{x}_{<d}, x_d)$) is generally intractable. On the other hand, if $q_\theta$ is a traditional autoregressive model represented as a product of normalized conditionals, then $\hat{J}_{CSM}(\theta; p)$ will be efficient to optimize, but the normalization constraint may limit expressivity. In the following, we propose a parameterization tailored for CSM training, where we represent a joint distribution directly in terms of $s_{\theta,d}(\mathbf{x}_{<d}, x_d), d = 1, \cdots D$ without normalization constraints.

# 4 Autoregressive conditional score models

We introduce a new class of probabilistic models, named *autoregressive conditional score models* (AR-CSM), defined as follows:

**Definition 1.** *An autoregressive conditional score model over $\mathbb{R}^D$ is a collection of $D$ functions $\hat{s}_d(\mathbf{x}_{<d}, x_d) : \mathbb{R}^{d-1} \times \mathbb{R} \to \mathbb{R}$, such that for all $d = 1, \cdots, D$:*

1. *For all $\mathbf{x}_{<d} \in \mathbb{R}^{d-1}$, there exists a function $\mathcal{E}_d(\mathbf{x}_{<d}, x_d) : \mathbb{R}^{d-1} \times \mathbb{R} \to \mathbb{R}$ such that $\frac{\partial}{\partial x_d}\mathcal{E}_d(\mathbf{x}_{<d}, x_d)$ exists, and $\frac{\partial}{\partial x_d}\mathcal{E}_d(\mathbf{x}_{<d}, x_d) = \hat{s}_d(\mathbf{x}_{<d}, x_d)$.*

2. *For all $\mathbf{x}_{<d} \in \mathbb{R}^{d-1}$, $Z_d(\mathbf{x}_{<d}) \triangleq \int e^{\mathcal{E}_d(\mathbf{x}_{<d}, x_d)}dx_d$ exists and is finite (i.e., the improper integral w.r.t. $x_d$ is convergent).*

Autoregressive conditional score models are an expressive family of probabilistic models for continuous data. In fact, there is a one-to-one mapping between the set of autoregressive conditional score models and a large set of probability densities over $\mathbb{R}^D$:

**Theorem 3.** *There is a one-to-one mapping between the set of autoregressive conditional score models over $\mathbb{R}^D$ and the set of probability density functions $q(\mathbf{x})$ fully supported over $\mathbb{R}^D$ such that $\frac{\partial}{\partial x_d} \log q(x_d|\mathbf{x}_{<d})$ exists for all $d$ and $\mathbf{x}_{<d} \in \mathbb{R}^{d-1}$. The mapping pairs conditional scores and densities such that*

$$\hat{s}_d(\mathbf{x}_{<d}, x_d) = \frac{\partial}{\partial x_d} \log q(x_d|\mathbf{x}_{<d})$$

The key advantage of this representation is that the functions in Definition 1 are easy to parameterize (e.g., using neural networks) as the requirements 1 and 2 are typically easy to enforce. In contrast with typical autoregressive models, we do not require the functions in Definition 1 to be normalized. Importantly, Theorem 3 does not hold for previous approaches that learn a single score function for the joint distribution [25, 24], since the score model $s_\theta : \mathbb{R}^D \to \mathbb{R}^D$ in their case is not necessarily the gradient of any underlying joint density. In contrast, AR-CSM *always* define a valid density through the mapping given by Theorem 3.

In the following, we discuss how to use deep neural networks to parameterize autoregressive conditional score models (AR-CSM) defined in Definition 1. To simplify notations, we hereafter use $\mathbf{x}$ to denote the arguments for $s_{\theta,d}$ and $s_d$ even when these functions depend on a subset of its dimensions.

## 4.1 Neural AR-CSM models

We propose to parameterize an AR-CSM based on existing autoregressive architectures for traditional (normalized) density models (*e.g.*, PixelCNN++ [21], MADE [4]). One important difference is that the output of standard autoregressive models at dimension $d$ depend only on $\mathbf{x}_{<d}$, yet we want the conditional score $s_{\theta,d}$ to also depend on $x_d$.

To fill this gap, we use standard autoregressive models to parameterize a "context vector" $\mathbf{c}_d \in \mathbb{R}^c$ ($c$ is fixed among all dimensions) that depends only on $\mathbf{x}_{<d}$, and then incorporate the dependency on $x_d$ by concatenating $\mathbf{c}_d$ and $x_d$ to get a $c+1$ dimensional vector $\mathbf{h}_d = [\mathbf{c}_d, x_d]$. Next, we feed $\mathbf{h}_d$ into another neural network which outputs the scalar $s_{\theta,d} \in \mathbb{R}$ to model the conditional score. The network's parameters are shared across all dimensions similar to [15]. Finally, we can compute $\frac{\partial}{\partial x_d} s_{\theta,d}(\mathbf{x})$ using automatic differentiation, and optimize the model directly with the CSM divergence.

Standard autoregressive models, such as PixelCNN++ and MADE, model the density with a prescribed probability density function (*e.g.*, a Gaussian density) parameterized by functions of $\mathbf{h}_d$. In contrast, we remove the normalizing constraints of these density functions and therefore able to capture stronger correlations among dimensions with more expressive architectures.

## 4.2 Inference and learning

To sample from an AR-CSM model, we use one dimensional Langevin dynamics to sample from each dimension in turn. Crucially, Langevin dynamics only need the score function to sample from a density [19, 6]. In our case, scores are simply the univariate derivatives given by the AR-CSM.

Specifically, we use $s_{\theta,1}(x_1)$ to obtain a sample $\overline{x}_1 \sim q_\theta(x_1)$, then use $s_{\theta,2}(\overline{x}_1, x_2)$ to sample from $\overline{x}_2 \sim q_\theta(x_2 \mid \overline{x}_1)$ and so forth. Compared to Langevin dynamics performed directly on a high dimensional space, one dimensional Langevin dynamics can converge faster under certain regularity conditions [20]. See Appendix C.3 for more details.

During training, we use the CSM divergence (see Eq. (5)) to train the model. To deal with data distributions supported on low-dimensional manifolds and the difficulty of score estimation in low data density regions, we use noise annealing similar to [24] with slight modifications: Instead of performing noise annealing as a whole, we perform noise annealing on each dimension individually. More details can be found in Appendix C.

# 5 Density estimation with AR-CSM

In this section, we first compare the optimization performance of CSM with two other variants of score matching: Denoising Score Matching (DSM) [28] and Sliced Score Matching (SSM) [25], and compare the training efficiency of CSM with Score Matching (SM) [9]. Our results show that CSM is more stable to optimize and more scalable to high dimensional data compared to the previous score matching methods. We then perform density estimation on 2-d synthetic datasets (see Appendix B) and three commonly used image datasets: MNIST, CIFAR-10 [12] and CelebA [13]. We further show that our method can also be applied to image denoising and anomaly detection, illustrating broad applicability of our method.

## 5.1 Comparison with other score matching methods

**Setup** To illustrate the scalability of CSM, we consider a simple setup of learning Gaussian distributions. We train an AR-CSM model with CSM and the other score matching methods on a fully connected network with 3 hidden layers. We use comparable number of parameters for all the methods to ensure fair comparison.

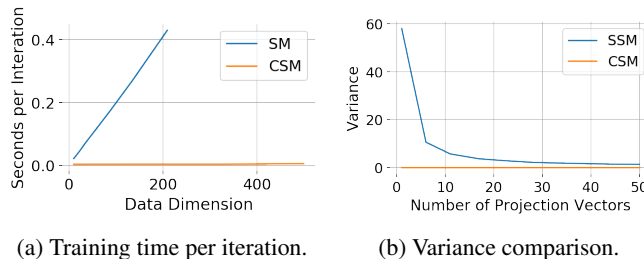

(a) Training time per iteration.      (b) Variance comparison.

Figure 1: Comparison with SSM and SM in terms of loss variance and computational efficiency.

**CSM vs. SM** In Figure 1a, we show the time per iteration of CSM versus the original score matching (SM) method [9] on multivariate Gaussians with different data dimensionality. We find that the training speed of SM degrades linearly as a function of the data dimensionality. Moreover, the memory required grows rapidly *w.r.t* the data dimension, which triggers memory error on 12 GB TITAN Xp GPU when the data dimension is approximately 200. On the other hand, for CSM, the time required stays stable as the data dimension increases due to parallelism, and no memory errors occurred throughout the experiments. As expected, traditional score matching (SM) does not scale as well as CSM for high dimensional data. Similar results on SM were also reported in [25].

**CSM vs. SSM** We compare CSM with Sliced Score Matching (SSM) [25], a recently proposed score matching variant, on learning a representative Gaussian $\mathcal{N}(0, 0.1^2 I)$ of dimension 100 in Figure 2 (2 rightmost panels). While CSM converges rapidly, SSM does not converge even after 20k iterations due to the large variance of random projections. We compare the variance of the two objectives in Figure 1b. In such a high-dimensional setting, SSM would require a large number of projection vectors for variance reduction, which requires extra computation and could be prohibitively expensive in practice. By contrast, CSM is a deterministic objective function that is more stable to optimize. This again suggests that CSM might be more suitable to be used in high-dimensional data settings compared to SSM.

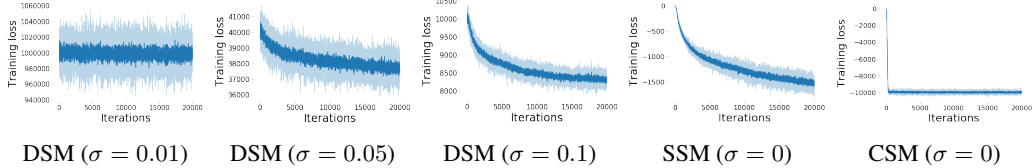

DSM ($\sigma = 0.01$)    DSM ($\sigma = 0.05$)    DSM ($\sigma = 0.1$)    SSM ($\sigma = 0$)    CSM ($\sigma = 0$)

Figure 2: Training losses for DSM, SSM and CSM on 100-d Gaussian distribution $\mathcal{N}(0, 0.1^2 I)$. Note the vertical axes are different across methods as they optimize different losses.

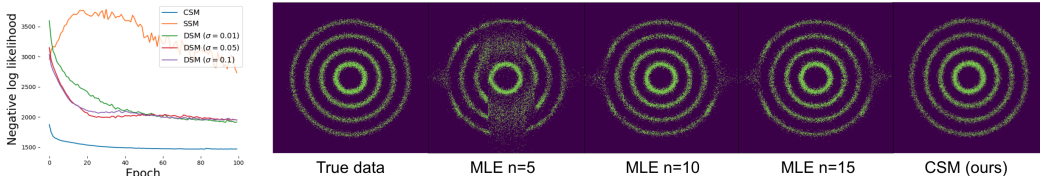

True data    MLE n=5    MLE n=10    MLE n=15    CSM (ours)

(a) MNIST negative log-likelihoods

(b) 2-d synthetic dataset samples from MADE MLE baselines with $n$ mixture of logistics and an AR-CSM model trained by CSM.

Figure 3: Negative log-likelihoods on MNIST and samples on a 2-d synthetic dataset.

**CSM vs. DSM**    Denoising score matching (DSM) [28] is perhaps the most scalable score matching alternative available, and has been applied to high dimensional score matching problems [24]. However, DSM estimates the score of the data distribution after it has been convolved with Gaussian noise with variance $\sigma^2 I$. In Figure 2, we use various noise levels $\sigma$ for DSM, and compare the performance of CSM with that of DSM. We observe that although DSM shows reasonable performance when $\sigma$ is sufficiently large, the training can fail to converge for small $\sigma$. In other words, for DSM, there exists a tradeoff between optimization performance and the bias introduced due to noise perturbation for the data. CSM on the other hand does not suffer from this problem, and converges faster than DSM.

**Likelihood comparison**    To better compare density estimation performance of DSM, SSM and CSM, we train a MADE [4] model with tractable likelihoods on MNIST, a more challenging data distribution, using the three variants of score matching objectives. We report the negative log-likelihoods in Figure 3a. The loss curves in Figure 3a align well with our previous discussion. For DSM, a smaller $\sigma$ introduces less bias, but also makes training slower to converge. For SSM, training convergence can be handicapped by the large variance due to random projections. In contrast, CSM can converge quickly without these difficulties. This clearly demonstrates the efficacy of CSM over the other score matching methods for density estimation.

## 5.2    Learning 2-d synthetic data distributions with AR-CSM

In this section, we focus on a 2-d synthetic data distribution (see Figure 3b). We compare the sample quality of an autoregressive model trained by maximum likelihood estimation (MLE) and an AR-CSM model trained by CSM. We use a MADE model with $n$ mixture of logistic components for the MLE baseline experiments. We also use a MADE model as the autoregressive architecture for the AR-CSM model. To show the effectiveness of our approach, we use strictly fewer parameters for the AR-CSM model than the baseline MLE model. Even with fewer parameters, the AR-CSM model trained with CSM is still able to generate better samples than the MLE baseline (see Figure 3b).

## 5.3    Learning high dimensional distributions over images with AR-CSM

In this section, we show that our method is also capable of modeling natural images. We focus on three image datasets, namely MNIST, CIFAR-10, and CelebA.

**Setup**    We select two existing autoregressive models — MADE [4] and PixelCNN++ [21], as the autoregressive architectures for AR-CSM. For all the experiments, we use a shallow fully connected network to transform the context vectors to the conditional scores for AR-CSM. Additional details can be found in Appendix C.

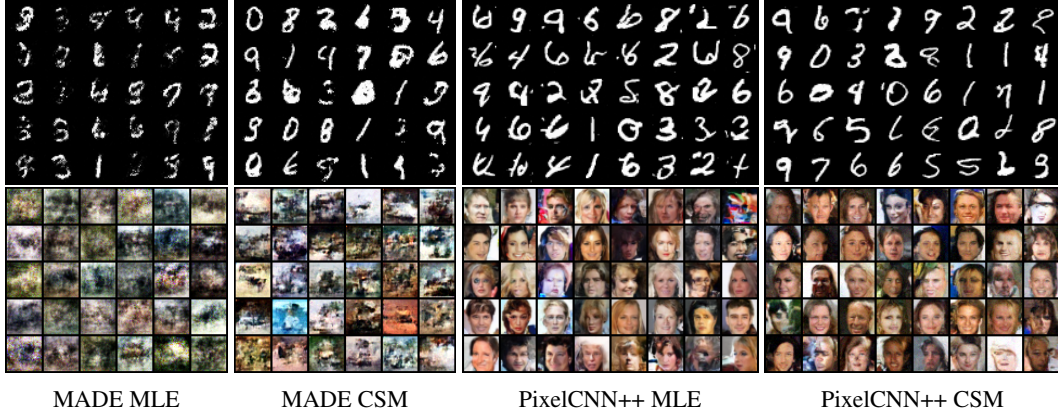

| MADE MLE | MADE CSM | PixelCNN++ MLE | PixelCNN++ CSM |

Figure 4: Samples from MADE and PixelCNN++ using MLE and CSM.

**Results** We compare the samples from AR-CSM with the ones from MADE and PixelCNN++ with similar autoregressive architectures but trained via maximum likelihood estimation. Our AR-CSM models have comparable number of parameters as the maximum-likelihood counterparts. We observe that the MADE model trained by CSM is able to generate sharper and higher quality samples than its maximum-likelihood counterpart using Gaussian densities (see Figure 4). For PixelCNN++, we observe more digit-like samples on MNIST, and less shifted colors on CIFAR-10 and CelebA than its maximum-likelihood counterpart using mixtures of logistics (see Figure 4). We provide more samples in Appendix C.

## 5.4 Image denoising with AR-CSM

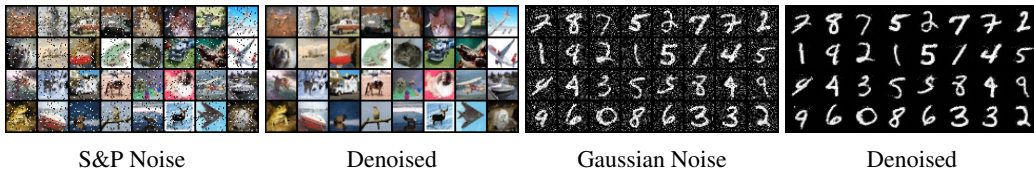

| S&P Noise | Denoised | Gaussian Noise | Denoised |

Figure 5: Salt and pepper denoising on CIFAR-10. Autoregressive single-step denoising on MNIST.

Besides image generation, AR-CSM can also be used for image denoising. In Figure 5, we apply $10\%$ "Salt and Pepper" noise to the images in CIFAR-10 test set and apply Langevin dynamics sampling to restore the images. We also show that AR-CSM can be used for single-step denoising [22, 28] and report the denoising results for MNIST, with noise level $\sigma = 0.6$ in the rescaled space in Figure 5. These results qualitatively demonstrate the effectiveness of AR-CSM for image denoising, showing that our models are sufficiently expressive to capture complex distributions and solve difficult tasks.

## 5.5 Out-of-distribution detection with AR-CSM

We show that the AR-CSM model can also be used for out-of-distribution (OOD) detection. In this task, the generative model is required to produce a statistic (*e.g.*, likelihood, energy) such that the outputs of in-distribution examples can be distinguished from those of the out-of-distribution examples. We find that $h_\theta(\mathbf{x}) \triangleq \sum_{d=1}^{D} s_{\theta,d}(\mathbf{x})$ is an effective statistic for OOD. In Tab. 1, we compare the Area Under

| Model | PixelCNN++ | GLOW | EBM | AR-CSM(Ours) |
|---|---|---|---|---|
| SVHN | 0.32 | 0.24 | 0.63 | **0.68** |
| Const Uniform | 0.0 | 0.0 | 0.30 | **0.57** |
| Uniform | **1.0** | **1.0** | **1.0** | 0.95 |
| Average | 0.44 | 0.41 | 0.64 | **0.73** |

Table 1: AUROC scores for models trained on CIFAR-10.

the Receiver-Operating Curve (AUROC) scores obtained by AR-CSM using $h_\theta(\mathbf{x})$ with the ones obtained by PixelCNN++ [21], Glow [10] and EBM [3] using relative log likelihoods. We use SVHN,

|  | MNIST (AIS) | | CelebA (FID) |
| --- | --- | --- | --- |
| Latent Dim | 8 | 16 | 32 |
| ELBO | 96.74 | 91.82 | 66.31 |
| Stein | 96.90 | 88.86 | 108.84 |
| Spectral | 96.85 | 88.76 | 121.51 |
| SSM | 95.61 | 88.44 | 62.50 |
| SSM-AR | 95.85 | 88.98 | 66.88 |
| CSM (Ours) | **95.02** | **88.42** | **62.20** |

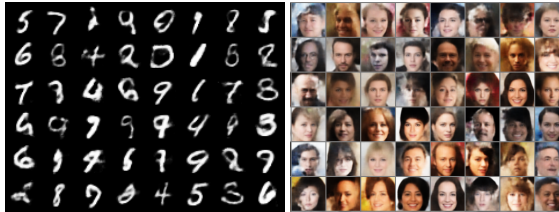

Table 2: VAE results on MNIST and CelebA.

Figure 6: CSM VAE MNIST and CelebA samples.

constant uniform and uniform as OOD distributions following [3]. We observe that our method can perform comparably or better than existing generative models.

## 6 VAE training with implicit encoders and CSM

In this section, we show that CSM can also be used to improve variational inference with implicit distributions [8]. Given a latent variable model $p_\theta(\mathbf{x}, \mathbf{z})$, where $\mathbf{x}$ is the observed variable and $\mathbf{z} \in \mathbb{R}^D$ is the latent variable, a Variational Auto-Encoder (VAE) [11] contains an encoder $q_\phi(\mathbf{z}|\mathbf{x})$ and a decoder $p_\theta(\mathbf{x}|\mathbf{z})$ that are jointly trained by maximizing the evidence lower bound (ELBO)

$$\mathbb{E}_{p_{data}(\mathbf{x})}[\mathbb{E}_{q_\phi(\mathbf{z}|\mathbf{x})} \log p_\theta(\mathbf{x}|\mathbf{z})p(\mathbf{z}) - \mathbb{E}_{q_\phi(\mathbf{z}|\mathbf{x})} \log q_\phi(\mathbf{z}|\mathbf{x})], \tag{6}$$

Typically, $q_\phi(\mathbf{z}|\mathbf{x})$ is chosen to be a simple *explicit* distribution such that the entropy term in Equation (6), $H(q_\phi(\cdot|\mathbf{x})) \triangleq -\mathbb{E}_{q_\phi(\mathbf{z}|\mathbf{x})}[\log q_\phi(\mathbf{z}|\mathbf{x})]$, is tractable. To increase model flexibility, we can parameterize the encoder using implicit distributions—distributions that can be sampled tractably but do not have tractable densities (*e.g.*, the generator of a GAN [5]). The challenge is that evaluating $H(q_\phi(\cdot|\mathbf{x}))$ and its gradient $\nabla_\phi H(q_\phi(\cdot|\mathbf{x})))$ becomes intractable.

Suppose $z_d \sim q_\phi(z_d|\mathbf{z}_{<d}, \mathbf{x})$ can be reparameterized as $g_{\phi,d}(\boldsymbol{\epsilon}_{\leq d}, \mathbf{x})$, where $g_{\phi,d}$ is a deterministic mapping and $\boldsymbol{\epsilon}$ is a $D$ dimensional random variable. We can write the gradient of the entropy with respect to $\phi$ as

$$\nabla_\phi H(q_\phi(\cdot|\mathbf{x}))) = -\sum_{d=1}^D \mathbb{E}_{p(\boldsymbol{\epsilon}_{<d})} \mathbb{E}_{p(\boldsymbol{\epsilon}_d)} \left[ \frac{\partial}{\partial z_d} \log q_\phi(z_d|\mathbf{z}_{<d}, \mathbf{x})|_{z_d = g_{\phi,d}(\boldsymbol{\epsilon}_{\leq d}, \mathbf{x})} \nabla_\phi g_{\phi,d}(\boldsymbol{\epsilon}_{\leq d}, \mathbf{x}) \right],$$

where $\nabla_\phi g_{\phi,d}(\boldsymbol{\epsilon}_{\leq d}, \mathbf{x})$ is usually easy to compute and $\frac{\partial}{\partial z_d} \log q_\phi(z_d|\mathbf{z}_{<d}, \mathbf{x})$ can be approximated by score estimation using CSM. We provide more details in Appendix D.

**Setup** We train VAEs using the proposed method on two image datasets – MNIST and CelebA. We follow the setup in [25] (see Appendix D.4) and compare our method with ELBO, and three other methods, namely SSM [25], Stein [26], and Spectral [23], that can be used to train implicit encoders [25]. Since SSM can also be used to train an AR-CSM model, we denote the AR-CSM model trained with SSM as SSM-AR. Following the settings in [25], we report the likelihoods estimated by AIS [16] for MNIST, and FID scores [7] for CelebA. We use the same decoder for all the methods, and encoders sharing similar architectures with slight yet necessary modifications. We provide more details in Appendix D.

**Results** We provide negative log-likelihoods (estimated by AIS) on MNIST and the FID scores on CelebA in Tab. 2. We observe that CSM is able to marginally outperform other methods in terms of the metrics we considered. We provide VAE samples for our method in Figure 6. Samples for the other methods can be found in Appendix E.

## 7 Related work

Likelihood-based deep generative models (*e.g.*, flow models, autoregressive models) have been widely used for modeling high dimensional data distributions. Although such models have achieved

promising results, they tend to have extra constraints which could limit the model performance. For instance, flow [2, 10] and autoregressive [27, 17] models require normalized densities, while variational auto-encoders (VAE) [11] need to use surrogate losses.

Unnormalized statistical models allow one to use more flexible networks, but require new training strategies. Several approaches have been proposed to train unnormalized statistical models, all with certain types of limitations. Ref. [3] proposes to use Langevin dynamics together with a sample replay buffer to train an energy based model, which requires more iterations over a deep neural network for sampling during training. Ref. [31] proposes a variational framework to train energy-based models by minimizing general $f$-divergences, which also requires expensive Langevin dynamics to obtain samples during training. Ref. [15] approximates the unnormalized density using importance sampling, which introduces bias during optimization and requires extra computation during training. There are other approaches that focus on modeling the log-likelihood gradients (scores) of the distributions. For instance, score matching (SM) [9] trains an unnormalized model by minimizing Fisher divergence, which introduces a new term that is expensive to compute for high dimensional data. Denoising score matching [28] is a variant of score matching that is fast to train. However, the performance of denoising score matching can be very sensitive to the perturbed noise distribution and heuristics have to be used to select the noise level in practice. Sliced score matching [25] approximates SM by projecting the scores onto random vectors. Although it can be used to train high dimensional data much more efficiently than SM, it provides a trade-off between computational complexity and variance introduced while approximating the SM objective. By contrast, CSM is a deterministic objective function that is efficient and stable to optimize.

## 8   Conclusion

We propose a divergence between distributions, named Composite Score Matching (CSM), which depends only on the derivatives of univariate log-conditionals (scores) of the model. Based on CSM divergence, we introduce a family of models dubbed AR-CSM, which allows us to expand the capacity of existing autoregressive likelihood-based models by removing the normalizing constraints of conditional distributions. Our experimental results demonstrate good performance on density estimation, data generation, image denoising, anomaly detection and training VAEs with implicit encoders. Despite the empirical success of AR-CSM, sampling from the model is relatively slow since each variable has to be sampled sequentially according to some order. It would be interesting to investigate methods that accelerate the sampling procedure in AR-CSMs, or consider more efficient variable orders that could be learned from data.

## Broader Impact

The main contribution of this paper is theoretical—a new divergence between distributions and a related class of generative models. We do not expect any direct impact on society. The models we trained using our approach and used in the experiments have been learned using classic dataset and have capabilities substantially similar to existing models (GANs, autoregressive models, flow models): generating images, anomaly detection, denoising. As with other technologies, these capabilities can have both positive and negative impact, depending on their use. For example, anomaly detection can be used to increase safety, but also possibly for surveillance. Similarly, generating images can be used to enable new art but also in malicious ways.

## Acknowledgments and Disclosure of Funding

This research was supported by TRI, Amazon AWS, NSF (#1651565, #1522054, #1733686), ONR (N00014-19-1-2145), AFOSR (FA9550-19-1-0024), and FLI.

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
