[Supplementary Material]

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

# A Proofs

## A.1 Regularity conditions

The following regularity conditions are needed for identifiability and integration by parts.

We assume that for every $\mathbf{x}_{<d}$ and for any $\theta$

1. $\frac{\partial}{\partial x_d} \log p(x_d|\mathbf{x}_{<d})$ and $\frac{\partial}{\partial x_d} \log q_\theta(x_d|\mathbf{x}_{<d})$ are continuously differentiable over $\mathbb{R}$.

2. $\mathbb{E}_{x_d \sim p(x_d|\mathbf{x}_{<d})}[\left(\frac{\partial \log p(x_d|\mathbf{x}_{<d})}{\partial x_d}\right)^2]$ and $\mathbb{E}_{x_d \sim p(x_d|\mathbf{x}_{<d})}[\left(\frac{\partial \log q_\theta(x_d|\mathbf{x}_{<d})}{\partial x_d}\right)^2]$ are finite.

3. $\lim_{|x_d| \to \infty} p(x_d|\mathbf{x}_{<d}) \frac{\partial \log q_\theta(x_d|\mathbf{x}_{<d})}{\partial x_d} = 0$.

## A.2 Proof of Theorem 1 (See page 3)

**Theorem 1** (CSM Divergence). $L_{CSM}(q_\theta, p)$ *vanishes if and only if* $q_\theta(\mathbf{x}) = p(\mathbf{x})$ *a.e.*

*Proof.* It is known that the Fisher divergence

$$L(q_\theta; p) = \frac{1}{2} \mathbb{E}_p \left[ \|\nabla_{\mathbf{x}} \log p(\mathbf{x}) - \nabla_{\mathbf{x}} \log q_\theta(\mathbf{x})\|_2^2 \right] \tag{7}$$

is a strictly proper scoring rule, and $L(q_\theta; p) \geq 0$ and vanishes if and only if $q_\theta = p$ almost everywhere [9].

Recall

$$L_{CSM}(q_\theta; p) = \frac{1}{2} \sum_{d=1}^{D} \mathbb{E}_{p(\mathbf{x}_{<d})} \mathbb{E}_{p(x_d|\mathbf{x}_{<d})} \left[ (s_d(\mathbf{x}) - s_{\theta,d}(\mathbf{x}))^2 \right]. \tag{8}$$

which we can rewrite as

$$L_{CSM}(q_\theta; p) = \sum_{d=1}^{D} \mathbb{E}_{p(\mathbf{x}_{<d})} \left[ L\left(q_\theta(x_d \mid \mathbf{x}_{<d}); p(x_d \mid \mathbf{x}_{<d})\right) \right]. \tag{9}$$

When $q_\theta = p$ almost everywhere, we have

$$q_\theta(\mathbf{x}_{\leq d}) = q_\theta(\mathbf{x}_{<d})q_\theta(x_d \mid \mathbf{x}_{<d}) = p(\mathbf{x}_{\leq d}) = p(\mathbf{x}_{<d})p(x_d \mid \mathbf{x}_{<d}) = q_\theta(\mathbf{x}_{<d})p(x_d \mid \mathbf{x}_{<d}) \quad a.e. \tag{10}$$

Let $\mathcal{A} \triangleq \{\mathbf{x}_{<d} \mid q_\theta(\mathbf{x}_{<d}) > 0\}$. We first observe that when $\mathbf{x}_{<d} \in \mathcal{A}$, Eq. equation 10 implies that $q_\theta(x_d \mid \mathbf{x}_{<d}) = p(x_d \mid \mathbf{x}_{<d})$ *a.e*, and subsequently, $(s_d(\mathbf{x}) - s_{\theta,d}(\mathbf{x}))^2 = 0$ *a.e.* Therefore

$$
\begin{aligned}
L_{CSM}(q_\theta; p) &= \frac{1}{2} \sum_{d=1}^{D} \mathbb{E}_{p(\mathbf{x}_{<d})} \mathbb{E}_{p(x_d|\mathbf{x}_{<d})} \left[ (s_d(\mathbf{x}) - s_{\theta,d}(\mathbf{x}))^2 \right] \\
&= \frac{1}{2} \sum_{d=1}^{D} \mathbb{E}_{p(\mathbf{x}_{<d})} \left[ \mathbb{I}[\mathbf{x}_{<d} \in \mathcal{A}] \mathbb{E}_{p(x_d|\mathbf{x}_{<d})} \left[ (s_d(\mathbf{x}) - s_{\theta,d}(\mathbf{x}))^2 \right] \right] \\
&= 0
\end{aligned}
$$

Now assume $L_{CSM}(q_\theta; p) = 0$. Because $L \geq 0$, $\mathbb{E}_{p(\mathbf{x}_{<d})} \left[ L\left(q_\theta(x_d \mid \mathbf{x}_{<d}); p(x_d \mid \mathbf{x}_{<d})\right) \right] \geq 0$ which means every term in the sum must be zero

$$\mathbb{E}_{p(\mathbf{x}_{<d})} \left[ L\left(q_\theta(x_d \mid \mathbf{x}_{<d}); p(x_d \mid \mathbf{x}_{<d})\right) \right] = 0 \ \forall d$$

and $L\left(q_\theta(x_d \mid \mathbf{x}_{<d}); p(x_d \mid \mathbf{x}_{<d})\right) = 0$ $p(\mathbf{x}_{<d})$-almost everywhere. Let's show that $q_\theta(\mathbf{x}_{\leq d}) = p(\mathbf{x}_{\leq d})$ almost everywhere using induction. When $d = 1$, $L\left(q_\theta(x_1); p(x_1)\right) = 0$ almost everywhere implies $q_\theta(x_1) = p(x_1)$ almost everywhere. Assume the hypothesis

holds when $d = k$, that is $q_\theta(\mathbf{x}_{\leq k}) = p(\mathbf{x}_{\leq k}))$ almost everywhere. Using the fact that $L\left(q_\theta(x_{k+1} \mid \mathbf{x}_{<k+1}); p(x_{k+1} \mid \mathbf{x}_{<k+1})\right) = 0$ $p(\mathbf{x}_{<k+1})$-almost everywhere (*i.e.*, $p(\mathbf{x}_{\leq k})$-almost everywhere), we have

$$q_\theta(x_{\leq k+1}) = q_\theta(x_{k+1} \mid \mathbf{x}_{<k+1})q_\theta(\mathbf{x}_{\leq k}) = p(x_{k+1} \mid \mathbf{x}_{<k+1})p(\mathbf{x}_{\leq k}) = p(\mathbf{x}_{\leq k+1}) \quad a.e.$$

Thus, the hypothesis holds when $d = k + 1$. By induction hypothesis, we have $q_\theta(\mathbf{x}_{\leq d}) = p(\mathbf{x}_{\leq d})$ $a.e.$ for any $d$. In particular,

$$q_\theta(\mathbf{x}) = q_\theta(\mathbf{x}_{\leq D}) = p(\mathbf{x}_{\leq D}) = p(\mathbf{x}) \quad a.e.$$

$\square$

### A.3  Proof of Theorem 2 (See page 3)

**Theorem 2** (Formal Statement). *$L_{CSM}(\theta; p) = J_{CSM}(\theta; p) + C$ where $C$ is a constant does not depend on $\theta$.*

*Proof.* Recall

$$L_{CSM}(q_\theta; p) = \frac{1}{2} \sum_{d=1}^{D} \mathbb{E}_{p(\mathbf{x}_{<d})} \mathbb{E}_{p(x_d \mid \mathbf{x}_{<d})} \left[ (s_d(\mathbf{x}) - s_{\theta,d}(\mathbf{x}))^2 \right]. \tag{11}$$

which we can rewrite as

$$L_{CSM}(q_\theta; p) = \sum_{d=1}^{D} \mathbb{E}_{p(\mathbf{x}_{<d})} \left[ L\left(q_\theta(x_d \mid \mathbf{x}_{<d}); p(x_d \mid \mathbf{x}_{<d})\right) \right]. \tag{12}$$

where $L(q_\theta; p)$ is the Fisher divergence.

Under the assumptions, we can use Theorem 1 from [9] which shows that $L(q_\theta; p) = J(\theta; p) + C$, where $C$ is a constant independent of $\theta$ and $J(q_\theta; p)$ is defined as below:

$$J(q_\theta; p) = \mathbb{E}_p \left[ \frac{1}{2} \|\nabla_\mathbf{x} \log q_\theta(\mathbf{x})\|_2^2 + \mathrm{tr}(\nabla_\mathbf{x}^2 \log q_\theta(\mathbf{x})) \right],$$

Substituting into equation 12 we get

$$L_{CSM}(q_\theta; p) = \sum_{d=1}^{D} \mathbb{E}_{p(\mathbf{x}_{<d})} \left[ J\left(q_\theta(x_d \mid \mathbf{x}_{<d}); p(x_d \mid \mathbf{x}_{<d})\right) + C(\mathbf{x}_{<d}) \right] \tag{13}$$

$$= \sum_{d=1}^{D} \mathbb{E}_{p(\mathbf{x}_{<d})} \left[ \mathbb{E}_{p(x_d \mid \mathbf{x}_{<d})} \left[ \frac{1}{2} s_{\theta,d}(\mathbf{x})^2 + \frac{\partial}{\partial x_d} s_{\theta,d}(\mathbf{x}) \right] + C(\mathbf{x}_{<d}) \right] \tag{14}$$

$$= \sum_{d=1}^{D} \mathbb{E}_{p(\mathbf{x}_{<d})} \left[ \mathbb{E}_{p(x_d \mid \mathbf{x}_{<d})} \left[ \frac{1}{2} s_{\theta,d}(\mathbf{x})^2 + \frac{\partial}{\partial x_d} s_{\theta,d}(\mathbf{x}) \right] \right] + C \tag{15}$$

$\square$

### A.4  Proof of Theorem 3

*Proof.* Let $\mathcal{Q}$ be the set of joint distributions that satisfies the condition in Theorem 3. Let $f$ be defined as:

$$f : \text{AR-CSM} \to \mathcal{Q}$$

$$f : \hat{s}(\mathbf{x}) = (\hat{s}_1(x_1), ..., \hat{s}_D(\mathbf{x}_{<D}, x_D)) \mapsto q(\mathbf{x}) := \prod_{d=1}^{D} \frac{e^{\mathcal{E}_d(\mathbf{x}_{<d}, x_d)}}{Z_d(\mathbf{x}_{<d})}$$

**Surjectivity**
Given $q \in \mathcal{Q}$, from the chain rule, we have

$$q(\mathbf{x}) = \prod_{d=1}^{D} q(x_d \mid \mathbf{x}_{<d}). \tag{16}$$

By assumption, we have $\frac{\partial}{\partial x_d} \log q(x_d|\mathbf{x}_{<d})$ exists for all $d$. Define $\hat{s}_d(\mathbf{x}_{<d}, x_d) = \frac{\partial}{\partial x_d} \log q(x_d|\mathbf{x}_{<d})$ for each $d$. We can check that $\mathcal{E}_d(\mathbf{x}_{<d}, x_d) = \log q(x_d|\mathbf{x}_{<d}) + C$, where $C$ is a constant. We also have $Z_d(\mathbf{x}_{<d}) = e^C \int q(x_d|\mathbf{x}_{<d}) dx_d = e^C$ exists. Thus, $s(\mathbf{x}) \in$ AR-CSM. On the other hand, we have

$$f(s(\mathbf{x})) = \prod_{d=1}^{D} \frac{e^{\mathcal{E}_d(\mathbf{x}_{<d}, x_d)}}{Z_d(\mathbf{x}_{<d})} \tag{17}$$

$$= \prod_{d=1}^{D} \frac{e^C \log q(x_d|\mathbf{x}_{<d})}{e^C} \tag{18}$$

$$= \prod_{d=1}^{D} \log q(x_d|\mathbf{x}_{<d}) \tag{19}$$

$$= q(\mathbf{x}) \tag{20}$$

Thus $s(\mathbf{x}) \in$ AR-CSM is a pre-image of $q(\mathbf{x})$ and $f$ is surjective.

**Injectivity**

Given $q \in \mathcal{Q}$, assume there exist $\hat{s}_1(\mathbf{x}), \hat{s}_2(\mathbf{x}) \in$ AR-CSM such that $f(\hat{s}_1(\mathbf{x})) = f(\hat{s}_2(\mathbf{x})) = q(\mathbf{x})$, we have

$$q(\mathbf{x}) = f(\hat{s}_i(\mathbf{x}))$$

$$\prod_{d=1}^{D} q(x_d|\mathbf{x}_{<d}) = \prod_{d=1}^{D} \frac{e^{\mathcal{E}_{i,d}(\mathbf{x}_{<d}, x_d)}}{Z_{i,d}(\mathbf{x}_{<d})},$$

where $i = 1, 2$.

**Lemma 1.** *For any $d = 1, ..., D$, we have $q(x_d|\mathbf{x}_{<d}) = \frac{e^{\mathcal{E}_{i,d}(\mathbf{x}_{<d}, x_d)}}{Z_{i,d}(\mathbf{x}_{<d})}$.*

*Proof.* Let's prove this argument using induction on $d$.
i) When $d = 1$, integrate equation 20 *w.r.t.* $x_D, ..., x_2$ sequentially, we have

$$\int ... \int \prod_{d=1}^{D} q(x_d|\mathbf{x}_{<d}) dx_D...dx_2 = \int ... \int \prod_{d=1}^{D} \frac{e^{\mathcal{E}_{i,d}(\mathbf{x}_{<d}, x_d)}}{Z_{i,d}(\mathbf{x}_{<d})} dx_D...dx_2$$

$$q(x_1) = \frac{e^{\mathcal{E}_{i,d}(x_1)}}{Z_{i,d}(x_1)}$$

Thus, the condition holds when $d = 1$.
ii) Assume the condition holds for any $d \leq k$, that is $q(x_d|\mathbf{x}_{<d}) = \frac{e^{\mathcal{E}_{i,d}(\mathbf{x}_{<d}, x_d)}}{Z_{i,d}(\mathbf{x}_{<d})}$ for any $d \leq k$. This implies

$$q(x_1, ..., x_k) = \prod_{d=1}^{k} q(x_d|\mathbf{x}_{<d}) = \prod_{d=1}^{k} \frac{e^{\mathcal{E}_{i,d}(\mathbf{x}_{<d}, x_d)}}{Z_{i,d}(\mathbf{x}_{<d})}$$

Similarly, integrating equation 20 *w.r.t.* $x_D, ..., x_{k+2}$ sequentially will give us

$$q(x_1, ..., x_{k+1}) = \prod_{d=1}^{k+1} \frac{e^{\mathcal{E}_{i,d}(\mathbf{x}_{<d}, x_d)}}{Z_{i,d}(\mathbf{x}_{<d})}.$$

Plugging in $q(x_1, ..., x_k) = \prod_{d=1}^{k} \frac{e^{\mathcal{E}_{i,d}(\mathbf{x}_{<d}, x_d)}}{Z_{i,d}(\mathbf{x}_{<d})}$ and use the fact that $q(x_1, ..., x_k) \neq 0$ (since $q$ has support equals to the entire space by assumption), we obtain $q(x_{k+1}|\mathbf{x}_{<k+1}) = \frac{e^{\mathcal{E}_{i,k+1}(\mathbf{x}_{<k+1}, x_{k+1})}}{Z_{i,k+1}(\mathbf{x}_{<k+1})}$. Thus, the hypothesis holds when $d = k + 1$.
iii) By induction hypothesis, the condition holds for all $d$. $\qquad\square$

From Lemma 1, we have

$$q(x_d|\mathbf{x}_{<d}) = \frac{e^{\mathcal{E}_{i,d}(\mathbf{x}_{<d})}}{Z_{i,d}(\mathbf{x}_{<d})}$$

$$\log q(x_d|\mathbf{x}_{<d}) = \mathcal{E}_{i,d}(\mathbf{x}_{<d}, x_d) - \log Z_{i,d}(\mathbf{x}_{<d}).$$

Taking the derivative *w.r.t* $x_d$ on both sides, since $\log Z_{i,d}(\mathbf{x}_{<d}, x_d)$ does not depend on $x_d$, we conclude that

$$\frac{\partial}{\partial x_d} \log q(x_d|\mathbf{x}_{<d}) = \frac{\partial}{\partial x_d} \mathcal{E}_{i,d}(\mathbf{x}_{<d}, x_d) = \hat{s}_{i,d}(\mathbf{x}_{<d}, x_d),$$

where $i = 1, 2$. This implies $\hat{s}_1(\mathbf{x}) = \hat{s}_2(\mathbf{x})$, and $f$ is injective. $\qquad\square$

# B   Data generation on 2D toy datasets

We perform density estimation on a couple two-dimensional synthetic distributions with various shapes and number of modes using our method. In Figure 7, we visualize the samples drawn from our model. We notice that the trained AR-CSM model can fit multi-modal distributions well.

Figure 7: Samples from 2D synthetic datasets. The first row: data distribution. The second row: samples from AR-CSM.

# C   Additional details of AR-CSM experiments

## C.1   More Samples

**MADE MLE**

(a) MNIST samples.　　　　　　(b) CIFAR-10 samples.

Figure 8: MADE MLE samples.

**MADE CSM**

(a) MNIST samples.　　　　　　(b) CIFAR-10 samples.

Figure 9: MADE CSM samples.

**PixelCNN++ MLE**

(a) MNIST samples.　　(b) CIFAR-10 samples.　　(c) CelebA samples.

Figure 10: PixelCNN++ MLE samples.

**PixelCNN++ CSM**

(a) MNIST samples.　　(b) CIFAR-10 samples.　　(c) CelebA samples.

Figure 11: PixelCNN++ CSM samples.

## C.2　Noise annealing

Figure 12: Conditional noise annealing at dimension $d$. The context $\hat{c}_d$ only depends on the pixels in front of $\hat{x}_d$ in $\hat{\mathbf{x}}$ (*i.e.* the green ones in $\hat{\mathbf{x}}$).

Training score-based generative modeling has been a challenging problem due to the manifold hypothesis and the existence of low data density regions in the data distribution. [24] shows the efficacy of noise annealing while addressing the above challenges. Based on their arguments, we adopt a noise annealing scheme for training one dimensional score matching. More specifically, we choose a positive geometric sequence $\{\sigma_i\}_{i=1}^{L}$ that satisfies $\frac{\sigma_1}{\sigma_2} = ... = \frac{\sigma_{L-1}}{\sigma_L} > 1$ to be our noise levels. Since at each dimension, we perform score matching *w.r.t.* $x_d$ given $\mathbf{x}_{<d}$, the previous

challenges apply to the one dimensional distribution $p(x_d|\mathbf{x}_{<d})$. We thus propose to perform noise annealing only on the scalar $x_d$. This process requires us to deal with $x_d$ and $\mathbf{x}_{<d}$ separately. For convenience, let us denote the input for the autoregressive model (context network) as $\hat{\mathbf{x}}$, and the scalar pixel that will be concatenated with the context vector as $\tilde{x}_d$. We decompose the training process into $L$ stages. At stage $i$, we choose $\sigma_i$ to be the noise level for $\tilde{x}_d$ and use the perturbed noise distribution $p_{\sigma_i}(\tilde{x}_d|x_d) = \mathcal{N}(\tilde{x}_d|x_d, \sigma_i^2)$ to obtain $\tilde{x}_d$. We use a shared noise level $\hat{\sigma}$ among all stages for $\hat{\mathbf{x}}$ and use a perturbed noise distribution $p_{\hat{\sigma}}(\hat{\mathbf{x}}|\mathbf{x}) = \mathcal{N}(\hat{\mathbf{x}}, |\mathbf{x}, \hat{\sigma}^2 I_D)$. We feed $\hat{\mathbf{x}}$ to the context network to obtain the context vector $\hat{\mathbf{c}}_d$ and concatenate it with $\tilde{x}_d$ to obtain $\mathbf{h}_d = [\hat{\mathbf{c}}_d, \tilde{x}_d]$, which is then fed into the score network to obtain conditional scores $s_{\theta,d}(\mathbf{x})$ (see Figure 12). At each stage, we train the network until convergence before moving on to the next stage. We denote the learned data distribution and the model distribution at stage $i$ for the $d$-th dimension as $p_{\sigma_i}(\tilde{x}_d|\hat{\mathbf{x}}_{<d})$ and $q_{\theta,\sigma_i}(\tilde{x}_d|\hat{\mathbf{x}}_{<d})$ respectively. As the perturbed noise for $\tilde{x}_d$ gradually decreases *w.r.t.* the stages, we call this process *conditional noise annealing*. For consistency, we want the distribution of $\hat{\mathbf{x}}$ to match the final state distribution of $\tilde{\mathbf{x}} = (\tilde{x}_1, ..., \tilde{x}_D)$, we thus choose $\hat{\sigma} = \sigma_D$ as the perturbed distribution for $\hat{\mathbf{x}}$ among all the stages.

### C.3 Inference with annealed autoregressive Langevin dynamics

To sample from the model, we can sample each dimension sequentially using one dimensional Langevin dynamics. For the $d$-th dimension, given a fixed step size $\epsilon > 0$ and an initial value $\tilde{x}_d^{[0]}$ drawn from a prior distribution $\pi(\mathbf{x})$ (*e.g.*, a standard normal distribution $\mathcal{N}(0,1)$), the one dimensional Langevin method recursively computes the following based on the already sampled previous pixels $\mathbf{x}_{<d}^{[T]}$

$$x_d^{[t]} = x_d^{[t-1]} + \frac{\epsilon}{2} \frac{\partial}{\partial x_d^{[t-1]}} \log p(x_d^{[t-1]}|\mathbf{x}_{<d}^{[T]}) + \sqrt{\epsilon}\mathbf{z}_t, \ t = 1, ..., T, \tag{21}$$

where $\mathbf{z}_t \sim \mathcal{N}(0, 1)$. When $\epsilon \to 0$ and $T \to \infty$, the distribution of $x_d^{[T]}$ matches $p(x_d|\mathbf{x}_{<d})$, in which case $x_d^{[T]}$ is an exact sample from $p(x_d|\mathbf{x}_{<d})$ under some regularity conditions [30]. Similar as [24], we can use annealed Langevin dynamics to speed up the mixing speed of one dimensional Langevin dynamics. Let $\epsilon_0 > 0$ be a prespecified constant scalar, we decompose the sampling process into $L$ stages for each $d$. At stage $i$, we run autoregressive Langavin dynamics to sample from $p_{\sigma_i}(\tilde{x}_d|\hat{\mathbf{x}}_{<d})$ using the model $q_{\theta,\sigma_i}(\tilde{x}_d|\hat{\mathbf{x}}_{<d})$ learned at the $i$-th stage of the training process. We define the anneal Langevin dynamics update rule as

$$\tilde{x}_d^{[t]} = \tilde{x}_d^{[t-1]} + \frac{\epsilon}{2} \frac{\partial}{\partial \tilde{x}_d^{[t-1]}} \log q_{\theta,\sigma_i}(\tilde{x}_d^{[t-1]}|\hat{\mathbf{x}}_{<d}^{[T]}) + \sqrt{\epsilon}\mathbf{z}_t, \ t = 1, ..., T, \tag{22}$$

where $\mathbf{z}_t \sim \mathcal{N}(0, 1)$. We choose the step size $\epsilon = \epsilon_0 \cdot \frac{\sigma_i^2}{\sigma_L^2}$ for the same reasoning as discussed in [24]. At stage $i > 1$, we set the initial state $\tilde{x}_d^{[0]}$ to be the final samples of the previous simulation at stage $i - 1$; and at stage one, we set the initial value $\tilde{x}_1^{[0]}$ to be random samples drawn from the prior distribution $\mathcal{N}(0, 1)$. For each dimension $d$, we start from stage one, repeat the anneal Langevin sampling process for $\tilde{x}_d$ until we reach stage $L$, in which case we have sampled the $d$-th component from our model. Compared to Langevin dynamics performed on a high dimensional space, one dimensional Langevin dynamics is shown to be able to converge faster under certain regularity conditions [20].

### C.4 Setup

For CelebA, we follow a similar setup as [24]: we first center-crop the images to $140 \times 140$ and then resize them to $32 \times 32$. All images are rescaled so that pixel values are located between $-1$ and $1$. We choose $L = 10$ different noise levels for $\{\sigma_i\}_{i=1}^L$. For MNIST, we use $\sigma_1 = 1.0$ and $\sigma_L = 0.04$, and $\sigma_1 = 0.2$ and $\sigma_L = 0.04$ are used for CIFAR-10 and CelebA. We notice that for the used image data, due to the rescaling, a Gaussian noise with $\sigma = 0.04$ is almost indistinguishable to human eyes. During sampling, we find $T = 20$ for MNIST and $T = 10$ for CIFAR-10 and CelebA work reasonably well for anneal autoregressive Langevin dynamics in practice. We select two existing autoregressive models, MADE [4] and PixelCNN++ [21], as the architectures for our autoregressive context network (AR-CN). For all the experiments, we use a shallow fully connected network as the

architecture for the conditional score network (CSN). The amount of parameters for this shallow fully connected network is almost negligible compared to the autoregressive context network. We train the models for 200 epochs in total, using Adam optimizer with learning rate 0.0002.

# D  Additional details of VAE experiments

## D.1  Background

Given a latent variable model $p(\mathbf{x}, \mathbf{z})$ where $\mathbf{x}$ is the observed variable and $\mathbf{z}$ is the latent variable, a VAE contains the following two parts: i) an encoder $q_\phi(\mathbf{z}|\mathbf{x})$ that models the conditional distribution of the latent variable given the observed data; and ii) a decoder $p_\theta(\mathbf{x}|\mathbf{z})$ that models the posterior distribution of the latent variable. In general, a VAE is trained by maximizing the evidence lower bound (ELBO):

$$\mathbb{E}_{p_{\text{data}}(\mathbf{x})}[\mathbb{E}_{q_\phi(\mathbf{z}|\mathbf{x})} \log p_\theta(\mathbf{x}|\mathbf{z})p(\mathbf{z}) - \mathbb{E}_{q_\phi(\mathbf{z}|\mathbf{x})} \log q_\phi(\mathbf{z}|\mathbf{x})]. \tag{23}$$

We refer to this traditional training method as "ELBO" throughout the discussion. In ELBO, $q_\phi(\mathbf{z}|\mathbf{x})$ is often chosen to be a simple distribution such that $H(q_\phi(\cdot|\mathbf{x})) \triangleq -\mathbb{E}_{q_\phi(\mathbf{z}|\mathbf{x})}[\log q_\phi(\mathbf{z}|\mathbf{x})]$ is tractable, which constraints the flexibility of an encoder.

## D.2  Training VAEs with implicit encoders

Instead of parameterizing $q_\phi(\mathbf{z}|\mathbf{x})$ directly as a normalized density function, we can parameterize the encoder using an implicit distribution, which removes the above constraints imposed on ELBO. We call such encoder an implicit encoder. Denote $H_d(q_\phi(\cdot|\mathbf{z}_{<d}, \mathbf{x})) \triangleq -\mathbb{E}_{q_\phi(z_d|\mathbf{z}_{<d}, \mathbf{x})}[\log q_\phi(z_d|\mathbf{z}_{<d}, \mathbf{x})]$, using the chain rule of entropy, we have

$$H(q_\phi(\cdot|\mathbf{x}))) = -\mathbb{E}_{q_\phi(\mathbf{z}|\mathbf{x})}[\log q_\phi(\mathbf{z}|\mathbf{x})] = \sum_{d=1}^{D} H_d(q_\phi(\cdot|\mathbf{z}_{<d}, \mathbf{x})). \tag{24}$$

Suppose $z_d \sim q_\phi(z_d|\mathbf{z}_{<d}, \mathbf{x})$ can be parameterized as $z_d = h_{\phi,d}(\epsilon_d, \mathbf{z}_{<d}, \mathbf{x})$, where $\epsilon_d$ is a simple one dimensional random variable independent of $\phi$ (*i.e.* a standard normal) and $h_{\phi,d}$ is a deterministic mapping depending on $\phi$ at dimension $d$. By plugging in $\mathbf{z}_{<d}$ into $h_{\phi,d}$ and using $z_d = h_{\phi,d}(\epsilon_d, \mathbf{z}_{<d}, \mathbf{x})$ recursively, we can show that $z_d$ can be reparametrized as $g_{\phi,d}(\boldsymbol{\epsilon}_{\leq d}, \mathbf{x})$, which is a deterministic mapping depending on $\phi$. This provides the following equality for the gradient of $H_d(q_\phi(\cdot|\mathbf{x}))$ *w.r.t.* $\phi$

$$\nabla_\phi H_d(q_\phi(\cdot|\mathbf{z}_{<d}, \mathbf{x})) \triangleq -\nabla_\phi \mathbb{E}_{q_\phi(z_d|\mathbf{z}_{<d}, \mathbf{x})}[\log q_\phi(z_d|\mathbf{z}_{<d}, \mathbf{x})] \tag{25}$$

$$= -\mathbb{E}_{p(\boldsymbol{\epsilon}_d)}[\frac{\partial}{\partial z_d} \log q_\phi(z_d|\mathbf{z}_{<d}, \mathbf{x})|_{z_d=g_{\phi,d}(\boldsymbol{\epsilon}_{\leq d}, \mathbf{x})} \nabla_\phi g_{\phi,d}(\boldsymbol{\epsilon}_{\leq d}, \mathbf{x})] \tag{26}$$

(See Appendix D.3). This implies

$$\nabla_\phi H(q_\phi(\cdot|\mathbf{x}))) = -\sum_{d=1}^{D} \mathbb{E}_{p(\boldsymbol{\epsilon}_{<d})}\mathbb{E}_{p(\boldsymbol{\epsilon}_d)}\left[\frac{\partial}{\partial z_d} \log q_\phi(z_d|\mathbf{z}_{<d}, \mathbf{x})|_{z_d=g_{\phi,d}(\boldsymbol{\epsilon}_{\leq d}, \mathbf{x})} \nabla_\phi g_{\phi,d}(\boldsymbol{\epsilon}_{\leq d}, \mathbf{x})\right],$$

Besides the aforementioned encoder and decoder, to train a VAE model using an implicit encoder, we introduce a third model: an AR-CSM model with parameter $\tilde{\phi}$ denoted as $s_{\tilde{\phi}}(\mathbf{z}|\mathbf{x})$ that is used to approximate the conditional score $s_{\phi,d}(\mathbf{z}|\mathbf{x}) \triangleq \frac{\partial}{\partial z_d} \log q_\phi(z_d|\mathbf{z}_{<d}, \mathbf{x})$ of the implicit encoder. During training, we draw i.i.d. samples $\{\mathbf{z}^{(1)}, ..., \mathbf{z}^{(N)}\}$ from the implicit encoder $q_\phi(\mathbf{z}|\mathbf{x})$ and use these samples to train $s_{\tilde{\phi}}(\mathbf{z}|\mathbf{x})$ to approximate the conditional scores of the encoder using CSM. After $s_{\tilde{\phi}}(\mathbf{z}|\mathbf{x})$ is updated, we use the $d$-th component of $s_{\tilde{\phi}}(\mathbf{z}|\mathbf{x})$, the approximation of $\frac{\partial}{\partial z_d} \log q_\phi(z_d|\mathbf{z}_{<d}, \mathbf{x})$, as the substitution for $\frac{\partial}{\partial z_d} \log q_\phi(z_d|\mathbf{z}_{<d}, \mathbf{x})$ in equation 26. To compute $\nabla_\phi H(q_\phi(\cdot|\mathbf{x}))$, we can detach the approximated conditional score $s_{\tilde{\phi}}(\mathbf{z}|\mathbf{x})$ so that the gradient of $H(q_\phi(\cdot|\mathbf{x}))$ could be approximated properly using PyTorch backpropagation. This provides us with a way to evaluate the gradient of Eq. equation 6 *w.r.t.* $\phi$, which can be used to update the implicit encoder $q_\phi(\mathbf{z}|\mathbf{x})$.

### D.3 Reparameterization

$$\nabla_\phi H_d(q_\phi(\cdot|\mathbf{z}_{<d},\mathbf{x})) \triangleq -\nabla_\phi \mathbb{E}_{q_\phi(z_d|\mathbf{z}_{<d},\mathbf{x})}[\log q_\phi(z_d|\mathbf{z}_{<d},\mathbf{x})] \tag{27}$$

$$= -\nabla_\phi \mathbb{E}_{p(\boldsymbol{\epsilon}_d)}[\log q_\phi(g_{\phi,d}(\boldsymbol{\epsilon}_{\leq d},\mathbf{x}))] \tag{28}$$

$$= -\mathbb{E}_{p(\boldsymbol{\epsilon}_d)}[\nabla_\phi \log q_\phi(g_{\phi,d}(\boldsymbol{\epsilon}_{\leq d},\mathbf{x}))] \tag{29}$$

$$= -\mathbb{E}_{p(\boldsymbol{\epsilon}_d)}\big[\frac{\partial}{\partial z_d}\log q_\phi(z_d|\mathbf{z}_{<d},\mathbf{x})|_{z_d=g_{\phi,d}(\boldsymbol{\epsilon}_{\leq d},\mathbf{x})}\nabla_\phi g_{\phi,d}(\boldsymbol{\epsilon}_{\leq d},\mathbf{x})\big]. \tag{30}$$

### D.4 Setup

For CelebA, we follow the setup in [25]. We first center-crop all images to a patch of $140 \times 140$, and then resize the image size to $64 \times 64$. For MNIST experiments, we use RMSProp optimizer with a learning rate of 0.001 for all methods except for the CSM experiments where we use learning rate of 0.0002 for the score estimator. On CelebA, we use RMSProp optimizer with a learning rate of 0.0001 for all methods except for the CSM experiments where we use a learning rate of 0.0002 for the score estimator.

# E   VAE with implicit encoders

## CSM

Figure 13: From left to right: VAE CSM MNIST samples with latent dimension 8, VAE CSM MNIST samples with latent dimension 16, VAE CSM CelebA samples with latent dimension 32.

## ELBO

Figure 14: From left to right: VAE ELBO MNIST samples with latent dimension 8, VAE ELBO MNIST samples with latent dimension 16, VAE ELBO CelebA samples with latent dimension 32.

## Stein

Figure 15: From left to right: VAE Stein MNIST samples with latent dimension 8, VAE Stein MNIST samples with latent dimension 16, VAE Stein CelebA samples with latent dimension 32.

**Spectral**

Figure 16: From left to right: VAE Spectral MNIST samples with latent dimension 8, VAE Spectral MNIST samples with latent dimension 16, VAE Spectral CelebA samples with latent dimension 32.

**SSM-AR**

Figure 17: From left to right: VAE SSM-AR MNIST samples with latent dimension 8, VAE SSM-AR MNIST samples with latent dimension 16, VAE SSM-AR CelebA samples with latent dimension 32.

**SSM**

Figure 18: From left to right: VAE SSM MNIST samples with latent dimension 8, VAE SSM MNIST samples with latent dimension 16, VAE SSM CelebA samples with latent dimension 32.