[Reviews · NeurIPS 2020]

Review 1

Summary and Contributions: This paper proposes Composite Score Matching (CSM), a scalable variant of the Score Matching (SM) algorithm. CSM decomposes the original formulation into multiple univariate SM problems, resulting in a parallelizable training procedure for unnormalized autoregressive models. The paper also introduces a new model class AR-CSM, which contains autoregressive models with unnormalized conditionals and expands the usual (normalized) autoregressive models studied before. The authors then experimentally show the scalability and training efficiency (in terms of number of steps) of CSM compared to existing related methods. ----- Post-rebuttal: The author feedback successfully addressed my concerns and also clarified some misunderstandings I had. I have thus increased my score to 6.

Strengths: THEORY - This work introduces a very clean extension of the original score matching algorithm specialized to autoregressive (AR) models. The theoretical results mirror those of the original Score Matching paper and are intuitive to understand. Also, their formulation naturally leads to a new model class that is strictly bigger than normalized AR models -- this is important because AR models currently achieve the best density estimation performance across various domains (e.g. images, audio, text). SIGNIFICANCE & RELEVANCE - Given the recent growing interest on unnormalized/energy-based models, combined with the impressive performance of AR models, the result of the paper is timely and relevant to the research community.

Weaknesses: EXPERIMENTS * Section 5.1, Figure 2 - Given that the vertical axes are on completely different scales, it's unclear what this comparison shows. Even though the loss curve for CSM plateaus more quickly than DSM, that doesn't necessarily imply that the trained model achieves better density estimation performance. * Section 5.2, Figure 3 - The qualitative comparison between PixelCNN++ MLE vs. CSM doesn't seem very convincing, particularly for CelebA sample quality comparison. The paper claims "less shifted colors" when using CSM, but there doesn't seem to be a noticeable difference difference between MLE and CSM for CelebA. - If CSM also trains a score network on top of h_d = [c_d, x_d], doesn't that have more parameters than the baseline model used to generate c_d (in which case the comparison wouldn't be fair because AR-CSM is parametrized by a strictly bigger network)? * Section 5.3 - Image denoising can be done by a wide range of generative models. So without any comparison to other methods nor quantitative metric (such as PSNR), the denoising results only seems to serve as a quick sanity check. A more thorough experiment would be necessary to demonstrate the the models trained under CSM are "sufficiently expressive to capture complex distributions and solve difficult tasks." * Section 6 - The NLL and FID improvements are very small. This again casts a doubt on whether the extra expressivity gained by allowing unnormalized models is actually helpful.

Correctness: - One of the strengths of AR-CSM mentioned in the paper is its expressivity compared to normalized AR models. I agree that traditional AR models constitute a smaller model class than AR-CSM due to normalization. But to what extent is this beneficial? It would be informative to see a toy example that clearly shows how the expressivity gained by AR-CSM leads to better likelihood or sample quality. - On the flip side, CSM can only be applied to a more limited function class (i.e. AR-CSM) compared to "black-box" methods such as SSM. It's unclear whether the benefits of CSM make up for this drawback (of having a more restricted model class compared to SSM) or not.

Clarity: The paper is clearly written and well-organized. I especially enjoyed reading the theory section, as the presented theorems are easy to follow and very intuitive. The justification behind using CSM was also nice too, except the corresponding experimental results were rather weak. Overall, the writing quality is high.

Relation to Prior Work: Yes. The paper covers several important related papers on topic -- namely Noise Conditional Score Networks (NCSN), Sliced Score Matching (SSM) and Denoising Score Matching (DSM). The presented idea is fundamentally different from the existing work as it focuses on autoregressive models in particular.

Reproducibility: Yes

Additional Feedback: * "CSM" seems refer to both "Composite Score Matching" and "Conditional Score Models". I found this to be slightly confusing.


Review 2

Summary and Contributions: The paper introduces Composite Score Matching, a new divergence between distributions based on the idea of Composite Score, Score Matching and autoregressive modelling. The authors use autoregressive decomposition of data and model distributions and apply score matching to a collection of one dimensional conditional scores thus constructing a composite score. The authors also theoretically justify this score as a proper learning objective and form a class of expressive autoregressive conditional score models, which are then implemented via autoregressive architectures like MADE and PixelCNN++. They demonstrate effectiveness of these models in a number of tasks, e.g. modelling images, density estimation, image denoising, OOD detection and increasing model flexibility in VAEs. === Post rebuttal and discussion update === I acknowledge the author rebuttal and the discussion, I maintained my score.

Strengths: Exploring new divergences for generative modelling is a relevant direction for the machine learning community. To my knowledge, applying score matching in the autoregressive setting has not been examined before in the literature. The proposed autoregressive conditional score models are also theoretically justified and empirically demonstrated to improve on the previous autoregressive models.

Weaknesses: I do not see major weaknesses right away, but it seems that ancestral sampling (consequently sampling each element) employed to sample from AR-CSM may take quite a time when data is very high-dimensional. It might also be the case that earlier samples x_{<d} should be drawn accurately to ensure the quality of subsequent samples x_{>d}.

Correctness: It seems to be fine, although detailed proofs are delayed to the supplementary material.

Clarity: Yes, the paper is well-written except for rare typos (e.g. exits — line 206, celebaA — line 267). Also some more comprehensible discussion is in the Appendix.

Relation to Prior Work: Yes, to my knowledge, the prior work is well discussed.

Reproducibility: Yes

Additional Feedback:


Review 3

Summary and Contributions: This paper proposed the autoregressive score matching model, which models joint distribution in terms of unnormalized scores to improve model capacity. In order to train the new model, the authors introduce a new divergence between distributions named Composite Score Matching (CSM) which only depends on the derivates of univariate scores, getting rid of computing the trace of the Hessian matrix. Concretely, CSM decomposes the score matching into individual conditionals, resulting a simpler optimization problem. Theoretical proof of convergence has been provided. Experiments on three benchmarks of images demonstrate that CSM enjoys more stable and efficient training compared with previous score matching methods including denoising score matching and sliced score matching. Moreover, CSM can also be applied to natural image generation, image denoising, out-of-distribution detection and can also be applied to model the posterior distribution in VAEs. === Post rebuttal and discussion update === I acknowledge the author rebuttal and the discussion, I maintained my score.

Strengths: The proposed CSM method is well-motivated, both theoretically and practically. Theoretically, CSM improves the flexibility of traditional autoregressive model by removing the normalized score constraints, though satisfying the property of exact density estimation. Moreover, it significantly improve the efficiency of score matching by decomposing it into conditionals at each autoregressive step. Theoretical convergence guarantee has been provided, which is highly appreciated. Practically, experimental results have illustrated the effectiveness of CSM on several tasks, and showing the efficiency of training compared with other score matching methods.

Weaknesses: NA

Correctness: The claims and empirical methodology are correct.

Clarity: The paper is well-written and easy to follow, given multiple formulations and theorems.

Relation to Prior Work: The related work is well summarized and the the relation and differences of this work compared with previous works have been discussed clearly.

Reproducibility: Yes

Additional Feedback: For the experiments in section 6, CSM is applied to model the posterior distribution in VAEs. Another popular approach to model the posterior distribution is using normalizing flows. Have you conducted experiments to compare CSM with normalizing flows?


Review 4

Summary and Contributions: This work first introduces composite score matching (CSM), a divergence between distributions, which only depends on the derivatives of univariate log-conditionals. Based the CSM, it introduces an extension of autoregressive models, which removes the normalization constraints for the univariate conditionals. The authors evaluate the proposed method on some simple density estimation and variational inference tasks. The main contribution of this paper is theoretical.

Strengths: pros: - The idea is natural but novel. - The proposed method seems to be technically sound.

Weaknesses: cons: - The empirical results are mostly on synthetic or small-scale dataset. - There could be some misconfigurations in experimental setup (see my detailed comments).

Correctness: I am concerned about the empirical study.

Clarity: Yes.

Relation to Prior Work: Yes.

Reproducibility: Yes

Additional Feedback: Detailed comments: - In Figure 1(a), the computational environment need to be specified. - In Figure 2, what're sigmas for SSM and CSM? - For image experiment, it would be much better to use the modern convolutional architecture instead of a shallow fully connected network. - I am confused about the results in Figure 3 and their implications. For example, PixelCNN++ MLE underperforms PixelCNN++ CSM in Figure 3 (at least on MINIST). However, In Appendix D5, the architecture of AR-CSM score network is quite different from PiexelCNN++. In particular, the number of parameters and model capacity really matter, as the PiexelCNN++ only has 40 filters and 1 residual block for MNIST and CIFAR-10. In other words, the constructed PixelCNN++ MLE could be a poor baseline here. ====post rebuttal update==== Thanks for clarifications. It addresses my major concerns, so I increase my score accordingly.

[Author Response · NeurIPS 2020]

We thank all reviewers for providing constructive feedback. We are glad that [R1] believes our work is a **timely and**
**relevant** contribution. We thank [R1, R2, R3, R4] for acknowledging the **theoretical contribution** of the paper (a new
**divergence** for learning unnormalized autoregressive (AR) models), and [R2, R3, R4] for appreciating the **novelty** and
**motivation** of our work. [R1, R4] raise questions about the configuration in the experiments. We believe this is due to a
misunderstanding on the architectures used and number of parameters.

**[R4] Q1: Model architectures for image experiments.** The PixelCNN++ baseline model is a deep network with
$>100$ layers. "ResNet" in appendix refers to a **group** of convolution blocks for each of the many gated ResNet layers
we use. We apologize for the confusion and will clarify this as well as upload the code. For comparison, our AR-CSM
model uses **exactly the same** AR model architecture (also with **convolutional architecture**) as the MLE baseline.
However, unlike the MLE baseline which passes the output of the AR model to a pre-specified **normalized** density
function (*e.g.* mixture of logistics), we pass to a score network (with $< 1\%$ parameters compared to the AR part) and
learn an **unnormalized** density function via the proposed CSM divergence. We provide additional experiments showing
that CSM can **outperform** MLE baselines even with **strictly less** parameters (see **[R1] Q1** MNIST and rings below).

**[R4] Q2: Clarification on experiments setups.** We run all the experiments using exactly the same setting on a 12 GB
TITAN Xp GPU. We briefly mention this at line 186. We will clarify this more in the revision. We use $\sigma = 0$ for both
CSM and SSM in Figure 2 as they already worked well without noise perturbation in this setting.

**[R1] Q1: Extra parameters introduced by score network.** The extra number of parameters from the score network is
almost **negligible** (*i.e.* $< 1\%$ compared to the autoregressive part). Empirically, we find that CSM is able to outperform
an MLE baseline even with **strictly less** parameters (including the score network) by generating better MNIST digit
samples (see MNIST samples below). We also provide a "rings" synthetic experiment where we use **strictly less**
parameters for the AR-CSM model than the baseline MLE model. We use a MADE architecture for the AR model and
$n$ mixtures of logistic components for the MLE experiments. Even with **strictly less** parameters, CSM is still able to
generate better samples than the MLE baseline (see rings figure below).

MLE (**more** parameters) / CSM (**less** parameters) | True data | MLE n=5 | MLE n=10 | MLE n=15 | CSM (ours)

25
**[R1] Q2: Figure 2 loss curves and advantage over SSM.** We use Figure 2 to provide insights into the training
challenges of DSM and SSM, and we do not intent to claim better density estimation from Figure 2. To compare density
estimation performance, we train a MADE model with tractable likelihood using the three score matching methods
on MNIST (a **more complicated** distribution than the one in Figure 2) and report the negative log likelihood (see the
figure above). The loss curves in the above figure match our discussion in Section 5.1. For DSM, a smaller $\sigma$ introduces
less bias, but also makes training slower to converge. SSM can introduce a **high variance** when approximating the
trace of the Hessian matrix. CSM, however, converges quickly. We believe this shows the efficacy of CSM over the
other score matching methods for density estimation.

**[R1] Q3: Whether expressivity gained by unnormalized density is helpful.** In Section 6 Table 2, **all** the methods
except for ELBO use unnormalized densities; CSM (unnormalized) outperforms ELBO (normalized) by a significant
amount in all the settings. We believe the expressivity provided by an unnormalized density is helpful.

**[R1] Q4: Less shifted color compared to baseline and denoising results.** Although
it is difficult to quantitatively measure "shifted color" in samples, we believe the samples
marked in blue (from baseline) have inconsistent "shifted colors". CSM samples, in
contrast, have more consistent colors according to human observers. We believe image
denoising is not a simple task, and while we do not claim SOTA results, Figure 4 shows
the capability of CSM to capture complex distributions. Our image results also show the effectiveness of CSM compared
to the previous approach for training unnormalized AR models (see Figure 6).

**[R2] Q1: Writing suggestions and ancestral sampling.** We thank Reviewer #2 for pointing out the typos and writing
suggestions. We will fix them in the revision. The quality of subsequent samples $x_{>d}$ does depend on earlier samples
$x_{<d}$. We find our sampling algorithm able to generate $x_{<d}$ that work reasonably well in practice.

**[R3] Q1: Comparison with normalizing flows.** We thank Reviewer #3 for the advice. Due to time constraint, we
only perform normalizing flow experiments on MNIST. We use flow models with comparable number of parameters as
CSM. The flow models obtain AIS scores of 95.69 and 88.91 for VAE experiments with latent dimension 8 and 16 on
MNIST. We notice that CSM **outperforms** the flow models with these settings.

[Meta-Review · NeurIPS 2020]

This paper gives a scalable variant of score matching for partition-function-free fitting of probabilistic models: the model is written as an autoregressive conditioning chain and score matching is applied separately to each factor in the chain. All four reviewers recommend acceptance. There is a consensus that the proposed approach is novel, theoretically well-motivated, empirically well-validated, and of broad interest. Initially, several reviewers voiced concerns and questions about the empirical evaluations. The author response addressed most of these and the reviewers raised their scores accordingly. Given this the AC recommends acceptance, but strongly encourages the authors to incorporate all of the reviewer feedback in the camera ready version.